# Effect of Metastable Intermolecular Composites on the Thermal Decomposition of Glycidyl Azide Polymer Energetic Thermoplastic Elastomer

**DOI:** 10.3390/polym16152107

**Published:** 2024-07-24

**Authors:** Chao Sang, Yunjun Luo

**Affiliations:** 1School of Chemistry and Chemical Engineering, Dezhou University, Dezhou 253023, China; 2Shandong Provincial Engineering Research Center of Organic Functional Materials and Green Low-Carbon Technology, Dezhou 253023, China; 3Shandong Provincial Key Laboratory of Monocrystalline Silicon Semiconductor Materials and Technology, College of Chemistry and Chemical Engineering, Dezhou University, Dezhou 253023, China; 4School of Materials Science and Engineering Technology, Beijing Institute of Technology, Beijing 100086, China

**Keywords:** metastable intermolecular composite, thermal decomposition, GAP-ETPE

## Abstract

Glycidyl azide polymer energetic thermoplastic elastomer (GAP-ETPE) has become a research hotspot due to its excellent comprehensive performance. In this paper, metastable intermolecular energetic nanocomposites (MICs) were prepared by a simple and safe method, and the catalytic performance for decomposition of GAP-ETPE was studied. An X-ray diffraction (XRD) analysis showed that the MICs exhibited specific crystal formation, which proved that the MICs were successfully prepared. Morphology, surface area, and pore structure analysis showed that the Al/copper ferrite and Al/Fe_2_O_3_ MICs had a large specific surface area mesoporous structure. The Al/CuO MICs did not have a mesoporous structure or a large surface area. The structure of MICs led to their different performance for the GAP-ETPE decomposition catalysis. The increase in specific surface area is a benefit of the catalytic performance. Due to the easier formation of complexes, MICs containing Cu have better catalytic performance for GAP-ETPE decomposition than those containing Fe. The conclusions of this study can provide a basis for the adjustment of the catalytic performance of MICs in GAP-ETPE propellants.

## 1. Introduction

Energetic thermoplastic elastomers (ETPEs) are used as solid propellant binders to impart advantages such as high energy, blunt sensitivity, low characteristic signal, and recyclability of propellants [1]. Energetic thermoplastic elastomers refer to thermoplastic elastomers containing energetic groups such as nitrate group (-ONO_2_), nitro group (-NO_2_), nitroamine group (-NNO_2_), azide group (-N_3_), and difluoroamine group (-NF_2_) [2]. Among them, azide energetic thermoplastic elastomers have attracted extensive attention due to their high heat release, no need for oxygen consumption during decomposition, and good compatibility with nitroamine explosives, among which polyazide glycidyl ether (GAP)-based ETPE is the representative. GAP-based ETPE (GAP-ETPE) has a soft-hard segment structure, in which the soft segment is partially or wholly composed of GAP, which gives ETPE high energy and thus increases the combustion rate of GAP-ETPE propellant, and the hard segment is composed of urethane segments, which gives ETPE high tensile strength (tensile strength is about 4.8 MPa, elongation at break is about 580% [3]), so GAP-ETPE propellants have become a research hotspot in thermoplastic elastomer propellants [4]. As the propellant binder, the thermal decomposition performance of GAP-ETPE will seriously affect the combustion performance of the propellant and then affect the thrust of the propellant to the rocket, so it is essential to catalyze its thermal decomposition. Research shows that the decomposition and combustion of GAP-ETPE can be catalyzed by transition metal oxides. And the catalysis to the binder (GAP-ETPE) can significantly improve the combustion performance of the propellant. Li found that CuO and Cr_2_Cu_2_O_5_ can drastically reduce the decomposition temperature of the azide group [5].

Metastable intermolecular composites (MICs) have short fuel-oxidizer diffusion distance; hence, they have a shorter ignition time, a higher burning speed, and a faster reaction speed than traditional thermites [6,7,8]. MICs are composed of oxidizer particles (such as Fe_2_O_3_, CuO, Bi_2_O_3_, etc.) and nano-sized metal fuel (mostly Al) [9,10,11,12], which are also termed nano-thermites, nanocomposite energetic materials, super thermites, metastable intermixed composites, etc. MICs are widely used in airbag ignition materials, propellants, aerospace equipment, pyrotechnics, pressure-mediated molecular transfer, and biomedical-related applications, both in civilian and military applications [13,14,15]. In addition to the excellent reactivity, MICs can also play the role of combustion catalyst when applied in solid propellants. The transition metal oxides such as CuO, Fe_2_O_3_ and Bi_2_O_3_ in the above-mentioned MICs can give MICs catalytic properties to solid propellant combustion [16,17,18,19]. At present, there are no studies on the effect of the transition metal oxides in MICs on the decomposition of GAP-ETPE.

In this work, Al/copper ferrites (include CuFe_2_O_4_, CuO and Fe_2_O_3_) MICs were prepared by a facile sol-gel reaction under mild conditions (low temperature calcination). We studied the morphology and structure changes of copper ferrites and Al/copper ferrites MICs, and the influence of these changes on their catalytic performance for GAP-ETPEE decomposition.

## 2. Experiment

### 2.1. Chemicals

Hexamethylene di-isocyanate (HMDI) from Bayer. Co. (Leverkusen, Germany). was used without any treatment. A glycidyl azide polymer (GAP; OH equivalent: 26.71 mg KOH/g) from Liming Research Institute of Chemical Industry (Luoyang, China) was used after vacuum drying for 2 h at 90 °C. A polymerization catalyst (dibutyltin dilaurate, DBTDL) from Beijing Chemical Plant (Beijing, China) was dissolved into dibutyl phthalate after using. Also, 1,4-butanediol (BDO) from Beijing Chemical Plant was vacuum dried for 4 h at 85 °C. One of the bonding agents, N-(2-cyanoethyl) diethanolamine (CBA), was made in our laboratory [20]. Acrylonitrile (Beijing Tong Guang Fine Chemicals Company, Beijing, China) was purified by vacuum distillation. Copper nitrate trihydrate (Cu(NO_3_)_2_·3H_2_O), diethanolamine, iron nitrate nonahydrate (Fe(NO_3_)_3_·9H_2_O), 1,2-propylene oxide, absolute ethanol, n-hexane, ethyl acetate, N,N-Dimethylformamide, which are all analytical grade, were purchased from Beijing Tong Guang Fine Chemicals Company (Beijing, China) and used without further purification.

### 2.2. Preparation of Samples

The whole preparation route of the MICs/GAP-ETPE samples is illustrated in Figure 1.

(1)Preparation of GAP-ETPE

The preparation of the glycidyl azide polymer energetic thermoplastic elastomer (GAP-ETPE) was achieved by the method reported in the literature [3]. The chain extenders were CBA (whose -CN group gives GAP-ETPE the molecular bonding function) and BDO. The number-average molecular weight (Mn) of GAP-ETPE was about 29,800 g·mol^−1^, and the ratio of hard segment and soft segment was 3:7. 

(2)Preparation of MICs

The Al/copper ferrite MICs was prepared via the method in the reference [21]. The scale of *n*(Cu) and *n*(Fe) was 1:4, and *n*(Al)/(*n*(Fe) + *n*(Cu)) was 3, which were proven to be the best ratio in our previous research. The products were labeled as CFA. As a comparison, Al/Fe_2_O_3_ and Al/CuO MICs were prepared by the same method. The same amount of Cu(NO_3_)_2_·3H_2_O (or Fe(NO_3_)_3_·9H_2_O) were replaced by Fe(NO_3_)_3_·9H_2_O (or Cu(NO_3_)_2_·3H_2_O). The products were labeled as FA and CA. Furthermore, without the position of Al, copper ferrite, Fe_2_O_3_, and CuO were prepared by the same method, which were labeled as CF, F, and C. The label and preparation information of the products are shown in Table 1.

(3)Preparation of MICs/GAP-ETPE

First, the GAP-ETPE was dissolved in tetrahydrofuran. Afterwards, the MICs powder was added to the solution at the percentage of 1% and sonicated for 30 min. The solvent was then evaporated in a vacuum oven at 60 °C for 6 h. Finally, the mixture was mixed using an open mill. The products prepared with F, C, CF, FA, CA, and CFA were, respectively, denoted as GF, GC, GCF, GFA, GCA, and GCFA. In particular, the sample without a catalyst was named G0.

### 2.3. Measurements and Characterizations

A S4800 cold field scanning electron microscope (SEM) (Japan’s Hitachi Corporation, Tokyo, Japan) was used to observe the morphology of the MICs at an accelerating voltage of 15.0 kV. An ASAP 2020 volumetric analyzer (Micromeritics Instrument Corporation, Norcross, GA, USA) was employed to investigate the surface area and pore structure of the MICs (degassed at 120 °C for at least 6 h). An X’Pert Pro MPD (PANalytical, Netherlands) diffractometer with monochromatic Cu Ka radiation (λ = 1.5406 Å) at 40 kV and 40 mA was used to carry out the X-ray diffraction (XRD) measurements of the MICs with a scanning speed of 0.01° s^−1^ and a step size of 0.01 from 10° to 90° (2θ). A METTLER TOLEDO TGA/DSC 1-Thermogravimetric Analyzer (Zurich, Switzerland) was used to measure the thermal performance of GAP-ETPE under the conditions of an ultrapure nitrogen atmosphere, an uncovered alumina ceramic crucible, a heating rate of 10 °C min^−1^, and a heating range of 30 °C to 600 °C.

## 3. Results and Discussion

### 3.1. Characters of MICs

#### 3.1.1. Crystal Forms

Phase investigations of the crystallized products were performed by XRD and the powder diffraction patterns were presented in Figure 2. As shown in Figure 2a, the diffraction pattern of C, F, and CF can be indexed to CuO (standard PDF card NO. 48-1548), γ-Fe_2_O_3_ crystal (standard PDF card NO. 25-1402) and CuFe_2_O_4_ (standard PDF card NO. 25-0283) [22]. Crystal defects in copper ferrite crystals are more than those in the pure iron oxide and copper oxide crystals. Crystal defects are beneficial to its catalytic function [23].

It can also be found from Figure 2 that both CA, FA, and CFA have the same set of additional diffraction peaks, indicating that the existence of Al (standard PDF card NO. 04-0787) in the products and demonstrate the sol-gel process could hold the structure of Al nanocrystalline particles. The diffraction peaks intensity of Al in CA are higher than those in FA and CFA, which indicates more Al is exposed on the outside of the oxide structure, which can also be proven in the following SEM images. The relative content of CuO, γ-Fe_2_O_3_, or CuFe_2_O_4_ reduces with the existence of Al, so their diffraction peak gradually weakens. The existence of n-Al does not change the crystal form of CuO, γ-Fe_2_O_3_, or CuFe_2_O_4_. This proves that the MICs were successfully prepared.

#### 3.1.2. Morphological Characterization

The surface morphologies of the MICs were observed by SEM, as shown in Figure 3. It can be seen from the figures that F and CF have a rough surface, while the surface morphologies of C are relatively smooth. The sample surface became rougher as the Al was added. The particles with a diameter of about 90 nm, that is, n-Al particles, can be clearly observed in the structure of the CA. But the same particles were not observed in the FA and CA, which indicates that the n-Al particles exist inside the structure of Fe_2_O_3_ and CuFe_2_O_4_. And this composite form that exposes oxides to the outermost layer is conducive to increasing the contact probability between the catalyst and the catalyzed substance and improving the catalytic activity when used as a catalyst for other reactions.

#### 3.1.3. Specific Surface Area and Pore Volume

As shown in Figure 4, the nitrogen adsorption-desorption isotherms of both Fe_2_O_3_, copper ferrites, Al/copper ferrite MICs, and Al/Fe_2_O_3_ MICs belong to type IV, which means abundant mesopores in the structure of the samples. But the CuO and Al/CuO MICs show little mesopores. The measured specific surface areas (*S*_BET_) of samples are presented in Table 2. It can be seen from the table that the specific surface area of Al/copper ferrite MICs and Al/Fe_2_O_3_ MICs are similar and show similar specific surface area changes with the presence of n-Al. The specific surface area of the MICs rise sharply after the addition of n-Al. It is because, as part of the gel skeleton, the n-Al increased the porosity of the MICs.

Generally, materials which have large specific surface areas have an excellent catalytic performance. That is, they benefit from the large number of catalytic active sites. With the same n-Al content, the pore volume and specific surface area of the Al/copper ferrite MICs are comparable to those of the Al/Fe_2_O_3_ MICs.

### 3.2. Thermal Decomposition of GAP-ETPE

GAP-ETPE is a promising solid propellant binder and its thermal decomposition performance can significantly affect the combustion performance of propellants. The transition metal oxide contained in MICs can be used to catalyze the thermal decomposition of GAP-ETPE. With the help of the TG and DSC technique, we discussed the effects of MICs for the thermal decomposition of GAP-ETPE.

Figure 5a,b shows the TG and DTG curves of the MICs/GAP-ETPE, and Table 3 lists the relevant parameters. Figure 5c shows the thermal effects during the MICs/GAP-ETPE thermal decomposition, which are tested by DSC. In order to facilitate comparison, the data of pure GAP-ETPE are also presented in the above figures and tables. All of the samples exhibit three weight losses, corresponding to the three decomposition stages of GAP-ETPE [24]. The first weight loss stage is at 200~90 °C, and the weight loss mass fraction was about 30%, which was basically consistent with the mass fraction of the azide group (29.7%) in the GAP-based ETPE. At the same time, the first stage decomposition peak temperature (*T*_P1_) in DTG corresponds to the exothermic peak temperature (*T*_P_) in the DSC curves. Therefore, the first weight loss stage corresponds to the side-chain azide. The second weightless stage appears at 280~380 °C, which corresponds to the urethane segment formed by HMDI, CBA and BDO. The third thermal decomposition stage shows at 380~490 °C, which corresponds to the decomposition of the polyether backbone, is related to the literature report on the thermal decomposition of GAP-based ETPEs [20].

The thermal decomposition process of GAP-ETPE is shown in Figure 6. The first step of the thermal decomposition of azide groups is the breaking of the RN-N_2_ bond to generate nitrene, followed by the rearrangement of nitrogen to generate imine, and the release of N_2_ imine to generate NH_3_ through a H transfer and free radical transfer, or the breaking of the C-C bond to generate HCN, etc. The formation of NH_3_ is an exothermic reaction, and the formation of HCN is an endothermic reaction.

It can be seen from Figure 5 that the thermogravimetric curves of the different samples are similar. The thermal decomposition mechanism of the GAP-ETPE has not been changed by the addition of MICs, but each decomposition steps temperature was changed. The azide maximum decomposition rate temperature (*T*_P1_) of pure GA-ETPE is 268.9 °C. The *T*_P1_ declines obviously after the addition of MICs, indicating that the MICs significantly affect azide decomposition.

As shown in Figure 5, the *T*_P1_ of GC and GCA are lower, and those of GF and GFA are higher. The reason is that the activation center of CuO can form an activation complex with the azide group, which reduces the activation energy of the decomposition and promotes the cleavage of RN-N_2_ bonds to form nitrogen bins, and N_2_ is also released, which advances the peak temperature of the decomposition of the azide group. The ability of Fe_2_O_3_ to form an activation complex is lower than that of CuO. Due to the presence of the Cu element, the catalytic capacity of CF and CFA for the decomposition of azides is stronger than that of F and FA.

The *T*_P1_ of GFA is lower than that of GF, which is because the more catalytic active sites provided by the huge specific surface area of FA. However, GC and GCA show the opposite pattern. The reasons are obviously as follows: The presence of n-Al did not change the specific surface area of the Al/CuO MICs so that there are no more catalytic active sites in Al/CuO MICs. Furthermore, the self-heating thermal decomposition process of the GAP-ETPE will be slowed down by the “dilution” and heat-conduction effects of aluminum [25]. Because of the dilution to the Cu element, the catalytic ability of CFA is lower than CF. But thanks to a large increase in specific surface area (80.43 to 170.55 m^2^ g^−1^, see Table 2), the *T*_P1_ of GCFA is only 6.6 °C higher than that of GCF. In conclusion, the catalytic ability of the catalyst can be improved by controlling the morphology. Besides, there seems to be no obvious synergistic catalytic effect of Cu-Fe oxides for the decomposition of GAP-ETPE.

## 4. Conclusions

Novel MICs successfully prepared by simple and mild methods were employed for the thermal decomposition of GAP-ETPE. The prepared Al/copper ferrite and Al/Fe_2_O_3_ metastable intermolecular energetic nanocomposites had a mesoporous structure with a large specific surface area, and the specific surface area greatly increased with the addition of n-Al. The Al/CuO MICs did not have a mesoporous structure and the specific surface area is very small. The structure of MICs led to their different performance for the GAP-ETPE decomposition catalysis. The increase in the specific surface area can improve the catalytic performance of the GAP-ETPE decomposition. Due to the easier complex formation, Cu-containing MICs have a better catalytic performance for GAP-ETPE decomposition than Fe-containing MICs. The MICs can reduce the decomposition temperature of the GAP-ETPE azide group by up to 45.7 °C. The conclusion of this study can help promote the application of GAP-ETPE as the binder in solid propellant.

## Figures and Tables

**Figure 1 polymers-16-02107-f001:**
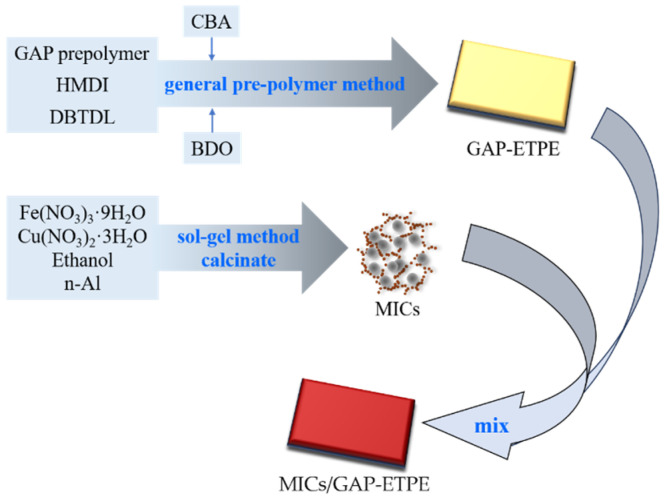
Schematic preparation process of MICs/GAP-ETPE.

**Figure 2 polymers-16-02107-f002:**
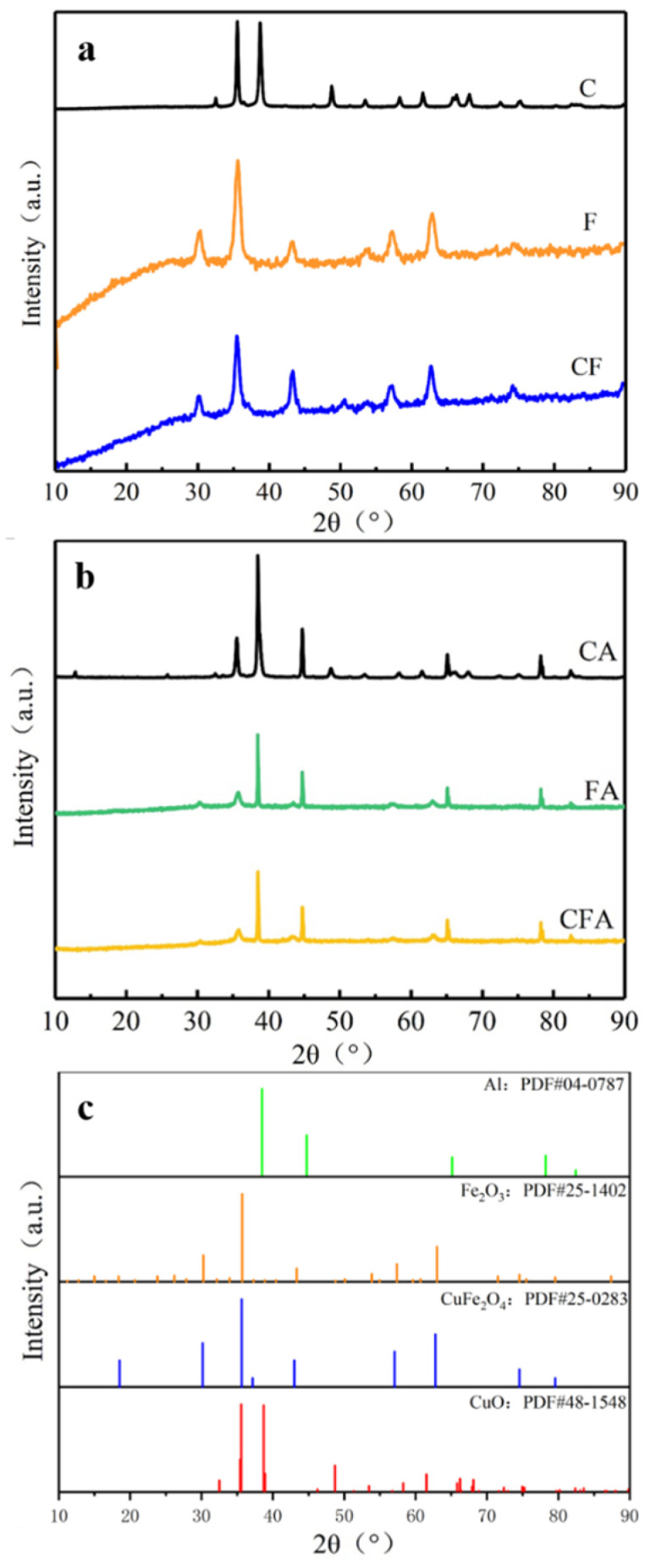
X-ray diffraction patterns of copper ferrites: (**a**) MICs (**b**) and PDF cards of Al, Fe_2_O_3_, CuFe_2_O_4_, CuO (**c**).

**Figure 3 polymers-16-02107-f003:**
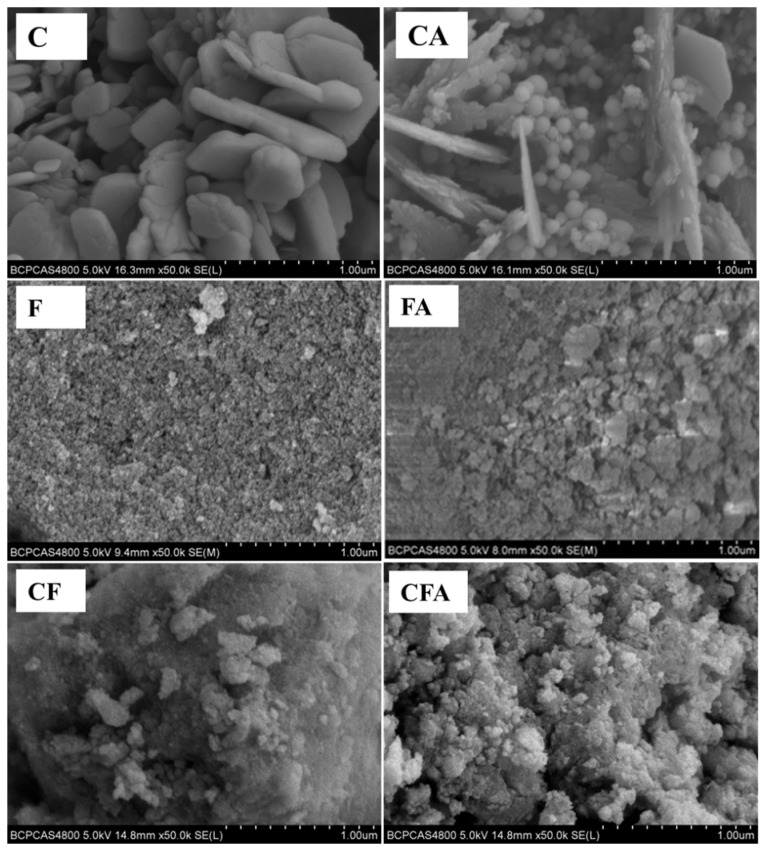
SEM images of Al/copper ferrite MICs.

**Figure 4 polymers-16-02107-f004:**
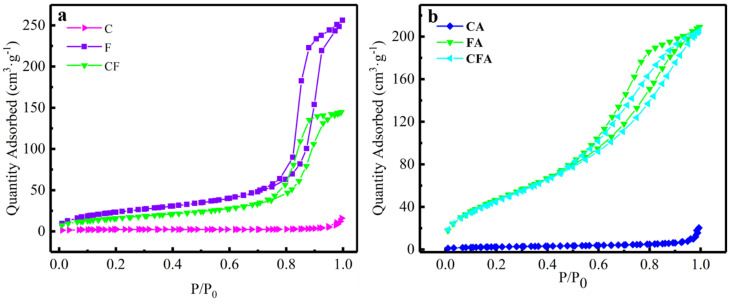
Nitrogen adsorption-desorption isotherms of copper ferrites (**a**) and MICs (**b**).

**Figure 5 polymers-16-02107-f005:**
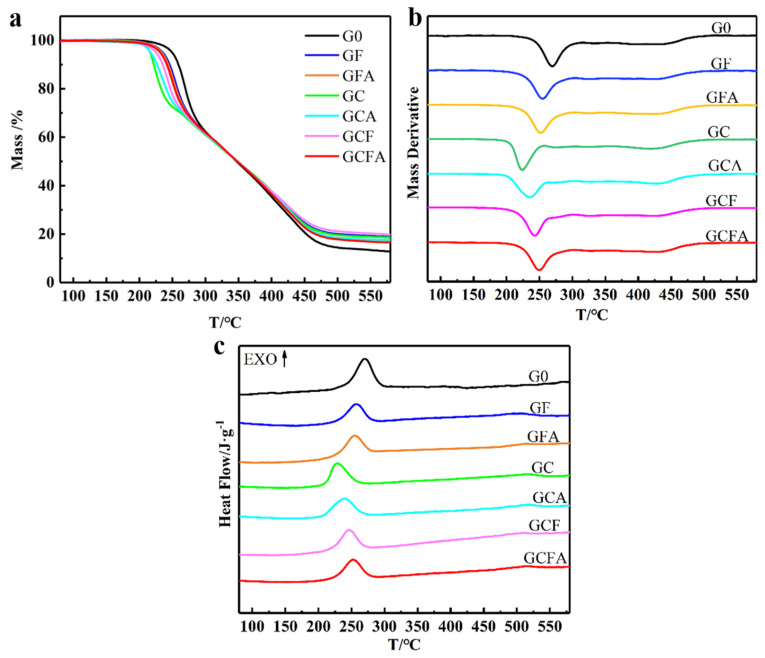
TG (**a**), DTG (**b**), and DSC (**c**) curves of MICs/GAP-ETPE.

**Figure 6 polymers-16-02107-f006:**
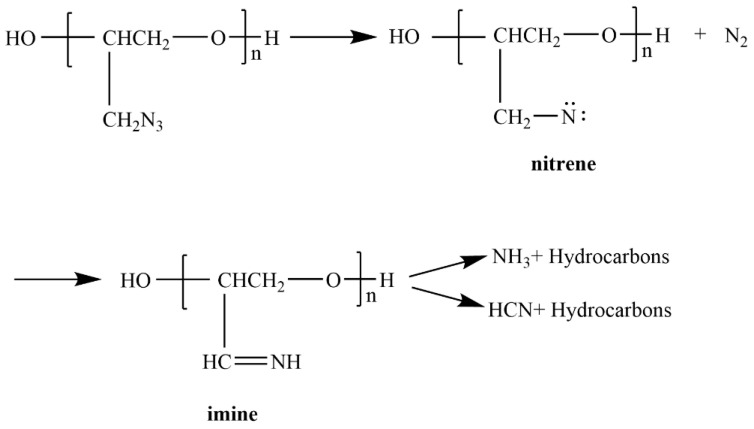
Thermal decomposition process of GAP-ETPE.

**Table 1 polymers-16-02107-t001:** The label and preparation information of the products.

Label	Product	Raw Material (mol)		
		Fe(NO_3_)_3_·9H_2_O	Cu(NO_3_)_2_·3H_2_O	Al
F	Fe_2_O_3_	0.060	0	0
C	CuO	0	0.060	0
CF	copper ferrite	0.012	0.048	0
FA	Al/Fe_2_O_3_ MICs	0.060	0	0.180
CA	Al/CuO MICs	0	0.060	0.180
CFA	Al/copper ferrite MICs	0.012	0.048	0.180

**Table 2 polymers-16-02107-t002:** Nitrogen adsorption-desorption isotherms parameters of MICs.

Samples	*S*_BET_/m^2^ g^−1^	*V*_tot_/cm^3^ g^−1^	*D*_ave_/nm
C	2.88	0.02	29.63
CA	6.39	0.03	15.92
F	87.92	0.39	17.07
FA	178.21	0.32	6.26
CF	80.43	0.26	11.77
CFA	170.55	0.31	6.39

Note: *S*_BET_ is the specific surface area calculated by the BET method, *V*_tot_ is the total pore volume, and *D*_ave_ is the average pore diameter.

**Table 3 polymers-16-02107-t003:** TG and DSC parameters of MICs/GAP-ETPE.

Samples	*T*_P1_ (°C)	*T*_P_ (°C)
G0	268.9	270.4
GC	223.2	227.5
GCA	235.5	239.9
GF	255.1	256.6
GFA	252.2	254.4
GCF	243.5	246.4
GCFA	250.1	252.9

## Data Availability

Data are contained within the article.

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
