# Peer review of "Effect of Metastable Intermolecular Composites on the Thermal Decomposition of Glycidyl Azide Polymer Energetic Thermoplastic Elastomer"

_polymers, 2024, doi:10.3390/polym16152107_

Round 1

Reviewer 1 Report

Comments and Suggestions for Authors

The revised manuscript can be accepted in its present form.

Author Response

Dear professor,

   Thank you for your pertinent comments on my manuscript.

   Best wishes.

Reviewer 2 Report

Comments and Suggestions for Authors

The manuscript "Effect of metastable intermolecular composites on the thermal decomposition of glycidyl azide polymer energetic thermoplastic elastomer" shows formation of MICs with GAP-ETPE where MICs functionalized as catalyst. The manuscript needs mor clarity as well more analytical characterizations. The language needs improvements and the reviewer suggest to let read it of a native speaker as several sentence are difficult to understand.

1. The introduction needs more background information as example why such MICs need to be added and if having such polymers GAP-ETPE how does the general properties as Young's modulus and tensile strength effected. Where are the application for such composites.

2. The materials and method needs more information as a simple scheme would help how those are made (as example see former research: https://doi.org/10.1016/j.combustflame.2024.113447). 

3. Additional characterization such as FTIR and EDX with mapping should be added to verify homogeneous contribution of MICs in the composites. Please add those. The organization of Figures as example if you have same method should be fused more together as example Figure 1-3. please display those as Figure 1a-c. Same goes for Figure 5 and 6 in combination Figure 5a and b, as well Figure 7 and 8 to Figure 6a-c. A more compact manuscript is much better readable.

4. The BET measurements as showing for FA higher values in SBET and smaller pores size than CFA. This small differences is not discussed in the text. Why has FA better specific surface volume and why does the TG and DSC parameter in Table 3 showing slightly reduced temperature for GCFA?  The authors said in conclusions that the increase of specific surface area (expressed over SBET) has an effect in catalytical activity, should therefore GFA not being enhanced in that than GCFA? the authors used a lot of abbreviations then switching to different composition in conclusion. Please add the different name you gave the samples matching in conclusion. Its a bit difficult to understand.

Comments on the Quality of English Language

A native speaker should correct the English as there are many mistakes in formed sentences and gramma.

Author Response

Response to reviewer

Dear reviewer:

We are truly grateful to your positive comments and suggestions. Based on these comments and suggestions concerning our manuscript entitled “Effect of Metastable Intermolecular Composites on the Thermal Decomposition of Glycidyl Azide Polymer Energetic Thermoplastic Elastomer” (ID: polymers-2964130), we have made careful modifications on the original manuscript (This includes asking native English speakers to help modify the grammar.). All changes made to the text are highlighted. We hope the newly revised manuscript will meet your magazine’s standard. If you still find some errors or inappropriate expressions, please give us more suggestions, we will revise our manuscript again.

Below you will find our point-by-point responses to the reviewers’ comments / questions:

Comments 1:

The introduction needs more background information as example why such MICs need to be added and if having such polymers GAP-ETPE how does the general properties as Young's modulus and tensile strength effected. Where are the application for such composites.

Answer:

The introduction has been revised. The mechanical properties of GAP-ETPE are added. The application of GAP-ETPE in propellants and its importance in thermal decomposition are described.

Comments 2:

The materials and method needs more information as a simple scheme would help how those are made (as example see former research: https://doi.org/10.1016/j.combustflame.2024.113447).

Answer:

The whole preparation route of MICs/GAP-ETPE samples is illustrated in Fig. 1.

Comments 3:

Additional characterization such as FTIR and EDX with mapping should be added to verify homogeneous contribution of MICs in the composites. Please add those. The organization of Figures as example if you have same method should be fused more together as example Figure 1-3. please display those as Figure 1a-c. Same goes for Figure 5 and 6 in combination Figure 5a and b, as well Figure 7 and 8 to Figure 6a-c. A more compact manuscript is much better readable.

Answer:

Thank you for your advice. Since GAP-ETPE is an energetic material, it may cause hazards and damage to the instrument when it is tested, so we are sorry that we did not supplement the relevant tests. However, according to literature reports (Li X, Ge Z, Li Q, Li D, Zuo Y, Yan B, Luo Y. "Effect of Burning Rate Catalysts on the Thermal Decomposition Properties of GAP-based ETPE Energetic Thermoplastic Elastormer." Chinese Journal of Energetic Materials, 2016.DOI:10.11943/j.issn.1006-9941.2016.11.013), such catalysts can be uniformly dispersed in GAP-ETPE. In addition, the related images have been merged according to your comments.

Comments 4:

The BET measurements as showing for FA higher values in SBET and smaller pores size than CFA. This small difference is not discussed in the text. Why has FA better specific surface volume and why does the TG and DSC parameter in Table 3 showing slightly reduced temperature for GCFA? The authors said in conclusions that the increase of specific surface area (expressed over SBET) has an effect in catalytical activity, should therefore GFA not being enhanced in that than GCFA? The authors used a lot of abbreviations then switching to different composition in conclusion. Please add the different name you gave the samples matching in conclusion. Its a bit difficult to understand.

Answer:

In this paper, we aim to discuss the effect of MICs on the thermal decomposition of GAP-ETPE. The performance of MICs has been studied in our previous published papers, so it is not necessary to repeat it here. The catalytic action on the decomposition of GAP-ETPE is affected by two factors, one is the specific surface area of the catalyst, and the other is the type of catalyst. The larger the specific surface area, the better the catalytic performance. However, the catalytic performance of copper oxide is better than that of iron oxide. Therefore, the decomposition temperature of GCFA is lower than that of GFA. Table 1 has listed the corresponding information of sample abbreviations, and the name of MICs/GAP-ETPE corresponds to it. Adding additional tables would be verbose, so we don't think it's necessary to add additional instructions.

Reviewer 3 Report

Comments and Suggestions for Authors

The manuscript by S. Chao et al. describes on “Effect of Metastable Intermolecular Composites on the Thermal Decomposition of Glycidyl Azide Polymer Energetic Thermoplastic Elastomer”. Here, metastable intermolecular energetic nanocomposites (MICs) were prepared by simple and safety method, and the catalytic performance for decomposition of glycidal azide polymer energetic thermoplastic elastomer (GAP-ETPE) was studied. The successful synthesis of MICs was done, since X-ray diffraction (XRD) analysis showed that the desired product had specific crystal structure. Morphology analysis by SEM, surface area and pore structure analysis by adsorption isotherms showed that the Al/copper ferrite and Al/Fe2O3  MICs had a large specific surface area mesoporous structure. The Al/CuO MICs does not have a mesoporous structure and the specific surface area is very small. The structure of MICs leads to their different performance for GAP-ETPE decomposition catalysis. The increase of specific surface area can improve the catalytic performance of GAP-ETPE decomposition as studied by thermogravimetric analysis (TGA). The decomposition mechanisms were clearly explained by TGA and differential scanning calorimetry. Due to the easier formation of complexes, MICs containing Cu have better catalytic performance for GAP-ETPE decomposition than those containing Fe. The conclusions of this study can provide further adjustment of the catalytic performance of MICs in GAP-ETPE propellants. Therefore, I find the materials described herein that include 9 Figures and one Table worthy of publication in Polymers. However, these are additional comments that must addressed in the revised version of this manuscript.

1) Please rephrase the Conclusion so that it is different from the Abstract.

2) In Fig. 9 please correct the structure of nitrene. Please provide the mechanisms for the formation of ammonia and hydrocarbon as well as for the formation of HCN and hydrocarbon from the imine structure.   

3) Please remove the duplicate reference in the References section and change accordingly in main text.

4) Please include all authors’ names in all the references, i.e., remove et al. from the references.

5) Please proofread the manuscript including the references to eliminate any typos for better readability.       

Comments on the Quality of English Language

Quality of English Language is ok.

Author Response

Response to reviewer

Dear reviewer:

We are truly grateful to your positive comments and suggestions. Based on these comments and suggestions concerning our manuscript entitled “Effect of Metastable Intermolecular Composites on the Thermal Decomposition of Glycidyl Azide Polymer Energetic Thermoplastic Elastomer” (ID: polymers-2964130), we have made careful modifications on the original manuscript. All changes made to the text are highlighted. We hope the newly revised manuscript will meet your magazine’s standard. If you still find some errors or inappropriate expressions, please give us more suggestions, we will revise our manuscript again.

Below you will find our point-by-point responses to the reviewers’ comments / questions:

Comments 1:

Please rephrase the Conclusion so that it is different from the Abstract.

Answer:

We have rephrased the Conclusion.

Comments 2:

In Fig. 9 please correct the structure of nitrene. Please provide the mechanisms for the formation of ammonia and hydrocarbon as well as for the formation of HCN and hydrocarbon from the imine structure.

Answer:

The structure of nitrene has been corrected. As described in lines 210 to 213 of this article, imines are produced by H transfer and free radical recombination to NH3 or by C-C bond breaking to HCN. The reaction that produces NH3 is exothermic and is dominant at low temperatures. The reaction that produces HCN is endothermic and is dominant at high temperatures.

Comments 3:

Please remove the duplicate reference in the References section and change accordingly in main text.

Answer:

All references have been screened and revised..

Comments 4:

Please include all authors’ names in all the references, i.e., remove et al. from the references.

Answer:

All references have been screened and revised..

Comments 4:

Please proofread the manuscript including the references to eliminate any typos for better readability.

Answer:

The full text has been checked and some grammatical and spelling errors have been corrected.

Round 2

Reviewer 2 Report

Comments and Suggestions for Authors

The authors answered all questions and made changes in the revised manuscript. The manuscript now in publishable form